# Preparation of Monoclonal Antibody against Deoxynivalenol and Development of Immunoassays

**DOI:** 10.3390/toxins14080533

**Published:** 2022-08-03

**Authors:** Hoyda Elsir Mokhtar, Aidi Xu, Yang Xu, Mohamed Hassan Fadlalla, Shihua Wang

**Affiliations:** Fujian Key Laboratory of Pathogenic Fungi and Mycotoxins, and School of Life Sciences, Fujian Agriculture and Forestry University, Fuzhou 350002, China; elsirhoyda@gmail.com (H.E.M.); 1200514073@fafu.edu.cn (A.X.); xuyang1996@fafu.edu.cn (Y.X.); awnab3@gmail.com (M.H.F.)

**Keywords:** deoxynivalenol, monoclonal antibody, cell fusion, ELISA, immunochromatographic strips

## Abstract

Fusarium toxins are the largest group of mycotoxins, which contain more than 140 known secondary metabolites of fungi. Deoxynivalenol (DON) is one of the most important compounds of this class due to its high toxicity and its potential to harm mankind and animals and a widespread contaminant of agricultural commodities, such as wheat, corn, barley, oats, bread, and biscuits. Herein, a hybridoma cell 8G2 secreting mAb against DON was produced by fusing the splenocytes with a tumor cell line Sp2/0. The obtained mAb had a high affinity (2.39 × 10^9^ L/mol) to DON. An indirect competitive Enzyme-Linked Immunosorbent Assay (ic-ELISA) showed that the linear range for DON detection was 3.125–25 μg/mL, and the minimum inhibitory concentration (IC_50_) was 18.125 μg/mL with a limit of detection (LOD) of 7.875 μg/mL. A colloidal gold nanoparticle (AuNP) with 20 nm in diameter was synthesized for on-site detection of DON within 10 min with vLOD of 20 μg/mL. To improve the limit of detection, the gold nanoflower (AuNF) with a larger size (75 nm) was used to develop the AuNF-based strip with vLOD of 6.67 μg/mL. Compared to the vLOD of a convectional AuNP-based strip, the AuNF-based strip was three times lower. Herein, three immunoassay methods (ic-ELISA and AuNP/AuNF-based strips) were successfully developed, and these methods could be applied for the DON detection in agricultural products.

## 1. Introduction

Deoxynivalenol (DON) is a small molecule secondary metabolite, well known as vomitoxin, which is secreted by various *Fusarium* species [1]. The genus *Fusarium* produces diverse mycotoxins known as trichothecenes, and DON is one of the most important compounds of this class due to its high toxicity. DON is classified as a type III carcinogen [2]. Ingesting feed or food exposed to DON can lead to abdominal suffering, reduced feed or food consumption, depression, diarrhea, neurotoxic and immunotoxic effects, shock, and death, particularly in high doses [3] in both animals and humans [4,5]. Mycotoxin toxicity occurs at very low concentrations, therefore, sensitive and reliable methods for their detection are required [6]. Various kinds of instrument-based approaches for DON detection have been utilized, including High-Performance Liquid Chromatography (HPLC) [7], Liquid Chromatography connected with Mass Spectrometry (LC-MS) [4], Gas Chromatography (GC) [8], and Gas Chromatography connected with Mass Spectrometry (GC-MS) [9]. These instrument- based approaches are highly sensitive and appropriate for multiple mycotoxins detection [10]. However, the drawbacks include sophisticated instrumentation, personnel experience, time-consuming, and unsuitability for the rapid screening of numerous samples. Recently, pioneer technologies with high sensitivity have been developed for DON detection, including chemiluminescence enzyme immunoassays [11,12], Raman [13], microfluidic method [14], electrochemical immunosensors [15], and lateral-flow devices (LFD). Compared to the sophisticated instrument-based method, immunoassays, such as enzyme-linked immunosorbent assay (ELISA) and lateral flow immunoassay (LFIA), have been broadly utilized for mycotoxins detection owing to attractive advantages including high sensitivity, simplicity, low commercial cost, and rapidity [16]. The lateral flow test, or immunochromatographic assay (ICA), is a direct visible detection method via naked eyes [17]. In addition, convectional LFIA was developed by integrating different nanoparticles as signal reporters to enhance their performance [18,19] with small surface area of 20–30 nm in diameter, such as gold nanosphere (AuNS) [20]. Gold nanoparticles remain most popular reporters for ICA due to their characteristics, such as easy of synthesis and modification [18]. The traditional colloidal gold-based ICA that use 20–40 nm colloidal gold (AuNP) as signal reporters has low sensitivity given the insufficient brightness of AuNP [21]. Hierarchical structures of gold nanoparticles with large and rough surface areas are more favorable for the strong absorption of antibodies than spherical AuNP [22]. Gold nanoflower (AuNF) possesses high optical brightness and strong target binding affinity because of its multibranched structure and large specific surface area. Thus, ICA with AuNF reporters has better sensitivity than ICA with conventional AuNP. Hierarchical flowerlike gold nanoparticles (AuNF) with a larger diameter (70 mm) of surface area provide higher signal intensity than conventional AuNS [23,24]. Kim et al. established ICA to detect both DON and ZEA, with vLOD of 50 ng/mL [25]. Herein, high affinity and specific monoclonal antibody (mAb) against DON was successfully developed, and based on obtained mAb, three immunoassay methods including ic-ELISA, AuNP-based strip, and AuNF-based strip, which could be applied for DON detection in agricultural commodities, were successfully developed.

## 2. Results

### 2.1. Animal Immunization and Hybridoma Screening

Eight-week old female Balb/c mice were immunized by DON-BSA conjugate. After four immunizations, sera were collected from immunized mice individually from the tail vein. Antibody titer was measured by the iELISA method using the DON-KLH conjugate to coat the plate. It was found that two immunized mice with complete conjugate (DON-BSA) showed higher antibody titer against standard DON toxin. The splenic cells of the immunized mouse with the highest antibody titer were isolated and fused with a tumor cell line Sp2/0. The hybridoma clones specific to DON were screened by a limited dilution method. Finally, two positive clones 4A4 and 8G2 secreting specific anti-DON mAb were successfully obtained in the study.

### 2.2. Isotypes and Chromosome Count

The subclass of obtained antibodies secreted by positive clones was assayed via a commercial iso-typing kit (IgA, IgGI, IgG2a, IgG2b, IgG3, IgM,). It was found that 4A4 and 8G2 secreting anti-DON mAb belonged to the IgG1 subtype as shown in Figure 1a. In the previously published paper, it was reported that the chromosome count of splenocyte was 66 ± 4, and a tumor Sp2/0 was 39 ± 1 [26]. The chromosome count of positive clone 4A4 and 8G2 was 95 ± 5 and 103 ± 5, respectively, as shown in Figure 1b. Hence, results of the chromosome count indicated that the number of chromosomes resulted from the fusion of the splenic cell of immunized mice and the tumor cell line Sp2/0.

### 2.3. Purification of Positive mAb

The positive hybridoma of interest (4A4 and 8G2) were injected into the mice abdomens for ascites production. The ascites fluid was collected and directly purified using caprylic acid/ammonium sulfate procedure. The purified antibodies were assayed using Sodium dodecyl sulfate-polyacrylamide gel electrophoresis (SDS-PAGE). Figure 2a,b show that a band size of heavy and light chains for 4A4 and 8G2 were 50 kDa and 27 kDa, respectively, corresponding to the molecular weight of IgG. Subsequently, the titer of the purified antibodies was measured by iELISA. Figure 2c showed the actual titer of the purified antibodies, indicating that a high titer of the antibody was obtained. The concentrations of the target antibodies were measured by BCA protein assay, and the result showed that the concentrations of secreted antibodies of 4A4 and 8G2 were 1.41 mg/mL and 1.70 mg/mL, respectively. 

### 2.4. Affinity and Speciality of the Positive mAb

The affinity of hybridoma clones (4A4 and 8G2) against DON was characterized by iELISA assay using DON-KLH with different concentrations (1.25–10 μg/mL) to coat the ELISA plate, and serial dilutions of the purified antibody were added into the plate. Analysis of the affinity constants (*Kaff*) of 4A4 and 8G2 against the DON antigen were obtained using Origin 9.1 for data analysis. The results showed that the mAb secreted by the two clones 4A4 and 8G2 were sensitive to DON, and the (*Kaff)* for 4A4 and 8G2 were 3.9 × 10^8^ L/mol and 2.39 × 10^9^ L/mol, respectively (Figure 3a,b). Specificity assay of hybridoma clones (4A4 and 8G2) was carried out according to the previous method [27]. Different toxins other than DON, such as penicillic acid (PA), fumonisin B_1_ (FB_1_)_,_ ochratoxins A (OTA), aflatoxin B_1_ (AFB_1_), ocadic acid (OA), and citrinin (CTN) were used as competitor antigens for the hybridoma clones (4A4 and 8G2). From the results, the anti-DON mAb only reacted to DON without cross-reactivity to other tested toxins (Figure 3c,d). Because the affinity binding of anti-DON mAb secreted by 8G2 was higher than that of the 4A4 cell line, 8G2 was selected for further experiments. 

### 2.5. Establishment of Standard Curve Based on ic-ELISA

The standard curve based on optimal conditions of the ic-ELISA method using the 8G2 anti-DON mAb was constructed. The relationship between DON concentration and its inhibition was analyzed using Origin 9.1 software (Figure 4a). The results showed the equation of logistic curve was y = 0.3372 + (0.78529 − 0.33772)/(1 + x/18671)^1.58782^, and the correlation coefficient (R^2^) was 0.98827. The linear equation was y = 2.27178 − 0.40137x, and the correlation coefficient (R^2^) was 0.95542 (Figure 4b). In this study, IC_50_ was 18.125 μg/mL. The linear range was 3.125–25 μg/mL, and LOD was 7.875 µg/mL.

### 2.6. Samples Detection

To assess the validity of the ic-ELISA method, the recovery test was performed, and four spike concentrations were selected (0.5, 5, 100, and 1000 ng/mL) for an artificially spiked DON-free corn sample. From results for intra assay, the recovery range was 87.80% to 96.53% with the average recovery of 91.46%, and the coefficient of variation (CV) range was 3.42% to 13.31%. The recovery range of inter assay was 92% to 96.61% with average recovery of 94.21%, and the CV range was 3.19% to 11.02% (Table 1). From the above results, the developed ELISA method in this study has a good stability and could be utilized for DON detection in real samples. 

### 2.7. Development of AuNP-Based Strip for Determination of DON

Colloidal gold (AuNP) about 20 nm in diameter was prepared using the standard citrate reduction procedure, and the red solution of AuNP was obtained which was easy to be seen by the naked eye. The structure of the AuNP-based strip is shown in Figure 5a. The DON-KLH conjugate, and the Goat anti-mouse IgG were sprayed onto a nitrocellulose (NC) membrane to form a test line (T line) and a control line (C line), respectively. If the DON-free sample was subjected to the strip sample pad, the AuNP probe directly bound the DON-KLH conjugate integrated on the NC, and a red line was shown on the T line (negative). If the samples contained DON, the AuNP probe reacted with the DON from the samples, which blocked the binding to the DON-KLH conjugates on the T line. Hence, the C line on the AuNP-based strip becomes lighter (positive). If both T and C lines showed a red line, it indicates a negative result. If the T line showed the red line but the C line did not or both the C and T line did not show the red line, it indicates an invalid test (Figure 5b). For specificity assessment of the strip, different toxins including ochratoxins A (OTA), fumonisin B1 (FB1), aflatoxin G1 (AFG1), ocadic acid (OA), and penicillic acid (PA) were individually dropped onto the developed AuNP strip sample pad. The AuNP-based strip test showed a red line on both T and C line except in the case of DON. This finding shows that the AuNP-based strip was highly specific to DON without any cross-reactivity (Figure 5c). For sensitivity assessment of the AuNP-based strip, different DON toxin concentrations were added individually (4 to 100 µg/mL) to the sample pads of different AuNP-based strips. From the result, the vLOD was 20 µg/mL (Figure 5d). Real samples (wheat flour, corn flour, barely, and oats) were purchased from a supermarket and analyzed for DON contamination. From the results, the developed color of the tested samples on the strip was the same as the negative control (PBS), indicating that these samples were DON-free (Figure 5e).

### 2.8. Preparation of AuNF Solution and AuNF-mAb Probe

To improve the detection limit of the conventional AuNP strip, the gold nanoflower (AuNF) with a large diameter of surface area was used [28], and the nanoparticles with different shape and size were also prepared [29]. For synthesis AuNF, the standard seeding growth method was used. It was reported that the reaction solution pH value together with the surface area of the AuNP influenced the antibody binding and adsorption. The prepared nanoflower solution was blue (Figure 6a). UV-vis absorption spectra showed that the peak of absorbance for the prepared AuNF solution was about 590 nm (Figure 6b). To synthesize the AuNF probe, the anti-DON mAb amount and AuNF reaction solution pH should be optimized. The optimal mount of anti-DON mAb for preparation of the AuNF probe was optimized to be 3 μL (6.12 µg) (Figure 6c). Moreover, the suitable pH of the AuNF reaction solution adjusted with 0.1M K_2_CO_3_ was considered to be 9 μL (pH about 6.8) (Figure 6d). Then the AuNF probe was synthesized under the optimal conditions. 

### 2.9. Construction and Characterization of the AuNF Strip

The AuNF strip just like the AuNP strip involved four components including sample pad (sample holder), absorption pad, and conjugate pad adhered to a nitrocellulose (NC) membrane. For sensitivity assessment of the AuNF strip, different concentrations of DON ranging from 4 to 100 µg/mL were added onto a sample pad of the developed AuNF strip. From the results, the blue on the strip (T line) became lighter with increased DON concentration (Figure 7A). Thus, the vLOD of the strip against DON was 6.67 μg/mL. For specificity evaluation of the developed AuNF strip, mycotoxins other than DON, such as AFB_1_, AFB_2,_ and AFG_1,_ were individually dropped into the sample pad, and the result showed blue lines on both T and C line except for the DON toxin (Figure 7B), indicating that the AuNF strip was very specific to DON without any cross-reactivity. Likewise, real samples including rice, wheat, and corn were investigated for DON contamination. From the result in Figure 7C, the strip showed the same result as the negative control PBS, indicating that these samples were DON-free.

## 3. Discussion

DON is a secondary metabolite small molecule which cannot induce mouse immune response, thus it requires carrier proteins to enhance mouse immunity to produce a specific monoclonal antibody. In this study, high antibody titers were induced by injecting the DON-BSA conjugate into the female Balb/c mouse, then the splenic cells were isolated from injected mice and fused with their partner tumor cell line Sp2/0. The successful cell fusions were achieved, and average fusion was 86.74% and the positive fusion rate was 48.95%. According to the previously reported studies, the high average of cell fusion rates was appropriate for hybridoma production [29,30]. The cell fusion and development of hybridoma cells in this study produced several positive cell clones, and their anti-DON antibody titers were determined by iELISA. The sub-class of obtained mAb in this study was an IgG1, which showed high specificity to DON. After screening, two positive clones 4A4 and 8G2 were selected according to their high titer and then were injected into the mice abdomens for ascities production. The ascitic fluid carrying specific anti-DON mAb was withdrawn and purified using caprylic/ammonium sulfate, and the SDS-PAGE method was utilized for determination of the heavy and light chain molecular weight of target antibodies. The results showed that the A4A and 8G2 mAbs were successfully purified, and the weight of the heavy and the light chain of 4A4 and 8G2 were 50 kDa and 27 kDa, respectively, which matches the weight of the heavy and the light chain of IgG in agreement with the previous study [30]. On the other hand, the specificity of the mAbs secreted by the A4A and 8G2 hybridoma cell was measured by ic-ELISA, and the results indicated that the anti-DON mAbs were specific to DON. In addition, an affinity constant value is essential for the antibody quality measuring the antigen–antibody reaction or binding quality, and the results showed that mAbs secreted by the two cell lines 4A4 and 8G2 were specific to DON, with an affinity constant 3.9 × 10^8^ L/mol and 2.39 × 10^9^ L/mol, respectively. According to the previously reported data, an antibody affinity within the range 10^7^–10^12^ L/mol indicated a good antibody potency [31]. Therefore, the anti-DON mAb secreted by 8G2 clone was specific to DON with good affinity, which was higher than that in the previously reported studies [30]. Therefore, 8G2 was selected for further experiments.

Based on the constructed standard curve, the IC_50_ was 18.125 μg/mL, LOD was 7.875 μg/mL, and the ic-ELISA linear ranged 3.125–25 μg/mL which was characterized as the concentration of DON inhibition from 20% to 80%. This finding indicated that the obtained anti-DON mAb secreted by 8G2 could be utilized to prepare an assay kit for DON detection. The result from intra-assay showed that the recovery ranged from 87.80% to 96.53%, the average recovery was 91.46%, and the coefficient of variation (CV) ranged from 3.42% to 13.31%. In the inter-assay, the recovery ranged from 92% to 96.61% with a recovery average 94.21%, and CV ranged from 3.19% to 11.02%. The CVs of intra- and inter-assay were both <14%. According to the Codex Alimentarius guide (CAC/GL 71-2009, the World Health Organization and the Agriculture Food Organization, Italy), the satisfactory CV for intra-laboratory assaying is <15%. Thus, this finding confirmed that the prepared ELISA method has very good stability. Our data recommend that the ELISA method prepared in the present study could meet the requirement of quantitative analytical methods. On the other hand, a rapid ICA strip was prepared for DON detection. At the same time, anti-DON mAbs secreted by 8G2 were conjugated with colloidal gold nanoparticles (AuNP). The significant advantage of the method applied is that the AuNP can be used as a rapid and on-side detection system without any instrument, providing an important way for DON detection. For sensitivity assessment, different concentrations of standard DON (4–100 µg/mL) were assayed. From the results, the vLOD of AuNP strip was 20 µg/mL. To improve the limit of detection, the gold nanoflower (AuNF) with a larger size (75 nm) was used to develop the AuNF-based strip with LOD 6.67 μg/mL. Compared to the vLOD of a convectional AuNP-based strip, the AuNF-based strip was three times lower. All these finding indicated that the developed immuno-assays are accurate, rapid, and sensitive and could be applied for the analysis of DON residues in agricultural products.

## 4. Conclusions

In this work, the high affinity of monoclonal antibody (mAb) against DON (2.39 × 10^9^ L/mol) secreted by the 8G2 cell line was obtained. Based on this mAb, three detection methods (ELISA, AuNP-based strip and AuNF-based strip) were developed. The linear range of ic-ELISA for DON detection was 3.125–25 μg/mL with LOD of 7.875 μg/mL, and the minimum inhibitory concentration (IC_50_) was 18.125 μg/mL. Moreover, mAb was labeled with colloidal gold nanoparticles (about 20 nm) to establish the conventional AuNP-based strip, and the vLOD of the strip test was 20 μg/mL. To improve the vLOD of the conventional AuNP-based strip, the hierarchical flowerlike structure AuNF was synthesized and utilized as the reporter to produce AuNF-based strip. The AuNF strip showed vLOD of 6.67 μg/mL. These results indicated the immunological methods developed in this work could be applied for DON detection in agricultural products.

## 5. Materials and Methods

### 5.1. Material and Reagents

Balb/c female mice were obtained from Wushi animal laboratory (Shanghai, China). Standard deoxynivalenol toxin (DON), Goat anti-mouse-peroxidase conjugate (IgG- HRP), Keyhole limpet haemocyanin (KLH), Ovalbumin (OVA), DON-KLH conjugate, Bovine serum albumin (BSA), DON-BSA conjugate, Chloroauric acid (HAuCl_4_•4H_2_O), Mouse monoclonal antibody isotyping kit (IgG1, IgG2a, IgG2b, IgG3, IgM, IgA), Fetal bovine serum (FBS), and RPMI 1640 were purchased from Sigma-Aldrich Chemical (St. Louis, MO, USA). The murine yeloma cell line Sp2/0 was stocked in liquid nitrogen at our laboratory. 

### 5.2. Immunization and Titer Determination

All the animal studies adhered to the Committee for Animal Ethics of Fujian Agriculture and Forestry University (FAFU) in China (C1017/23.Dec.2014). To obtain a highly sensitive as well as specific monoclonal antibody (mAb), the immunogen DON-BSA conjugate was used to immunize the 8-week old mice according to a previously published paper [32]. Equal volume of DON-BSA conjugate (100 µg) and Freund’s complete adjuvant were mixed before animal injection. In the first injection of immunization, the mixer was administrated intraperitoneally. Then, an equal volume of DON-BSA conjugate (100 µg) and Freund’s incomplete adjuvant were mixed for assist immunization, at 2-week intervals. After the fourth immunization, the blood was collected from the tail vein of the individual mouse and centrifuged at 5000 r/min for 35 min. Then the serum was stored at −20 °C for further use. The serum titer was measured by the iELISA method according to the previous publication [29]. 

### 5.3. Screening and Characterization of Hybridoma Cells

The positive hybridoma clones against DON were screened according to the published paper [33] with minor modifications. B-lymphocytes from the injected mouse were fused with its partner Sp2/0 myeloma cells at the ratio of 10:1 in the presence of PEG1450 [34]. Hybridoma cells were exhaustively cultivated into 96-well ELISA plates in the presence of feeder cells. Positive hybridoma were sub-cloned 3 times and isolated from the culture by limited dilution method. The positive hybridoma were cloned again for a second time and then hybridoma were characterized by the sub-class kit [35]. The chromosome number of hybridoma cells was measured using a Geimsa stain [27]. 

### 5.4. Purification and Characterization of mAb from Ascites

For ascites production, Balb/c mice were first injected with 500 μL of pristine intraperitoneally. One week later, the mice were injected with 8G2 hybridoma cells. At about 7 days, the produced ascites fluid was harvested. The ascites was purified by caprylic acid/ammonium sulfate precipitation methods. The purified mAb was assayed via SDS-PAGE [36], and antibody concentration was measured by BCA kit [37]. The affinity of target antibodies was determined based on the method previously described [24]. Different concentrations of the DON-KLH (1.25, 2.5, 5, and 10 μg/mL) were used to coat the ELISA microplate then incubated at 37 °C for 2 h, and a serial dilution of mAb in PBSM was dropped into the reaction. After washing and blocking steps, the HRP conjugated goat anti-mouse IgG (1:8000 dilutions) was finally added. The antibody affinity was evaluated using Origin 9.0 software [35]. For specificity assessment of the obtained mAbs, an ic-ELISA was conducted based on the previously published method [27], and different toxins, including ochratoxins A (OTA), fumonisin B_1_ (FB_1_), aflatoxin B_1_ (AFB_1_), citrinin (CTN), and penicillic acid (PA) with various concentrations (0.39, 0.78, 1.56, 3.13, 6.25, 12.5, 25, and 50 μg/mL) were applied as a competitor antigen to examine the cross reactivity of the obtained antibody [34].

### 5.5. Development of ic-ELISA and Standard Curve

Based on ic-ELISA and the optimal conditions, the standard curve was plotted [37,38]. In brief, the DON-KLH was diluted with coating buffer and used to coat the ELISA plate and incubated for 2 h at 37 °C to form a solidified antigen. Then the plate was washed and blocked with PBSM for 2 h at 37 °C. Then the blocking solution was removed, and the plate was triple-washed by PBST and then PBS, respectively. An equal volume of anti-DON mAb and standard DON toxin at different concentrations (0.39, 0.78, 1.56, 3.13, 6.25, 12.5, 25, and 50 μg/mL) were mixed and incubated at 37 °C for 30 min. After washing many times, the formed solidified antigen–antibody complex was retained on the microplate. The HRP conjugated goat anti-mouse IgG (1:8000) was dropped into the plate and incubated for 1 h at 37 °C. The remaining solidified antigen–antibody complex reacted with the HRP conjugated secondary antibody to form a new antigen-antibody complex with enzyme activity. After an extra wash step, the substrate TMB was dropped into the plate and incubated at 37 °C for 20 min. Eventually, the H_2_SO_4_ (2 mol/L) was added to terminate the reaction, and the absorbance at 450 nm was measured [24]. Finally, the standard curve was constructed, and the inhibition concentration of standard DON toxin in relation to anti-DON mAb was measured using data analysis software Origin Pro 9.1 (OriginLab, Northampton, MA, USA) [27]. An actual corn sample was spiked with four concentrations of DON standard toxin (0.5, 5, 100, and 1000 ng/mL). The corn sample was crushed and soaked in 10 mL of 70% methanol (*v*/*v*) and homogenized for 35 min. After extracted and centrifuged at 1200 r/min for 20 min, the obtained supernatant was assayed by ic-ELISA. The recovery test was conducted and measured based on the constructed standard curve, and the coefficient variation (CV) together with the recovery test was measured in triplicate.

### 5.6. Construction and Characterization of AuNP-Based Strip 

A colloidal gold particle (AuNP) was prepared by Masinde’s methods [39]. The amount of antibody and optimal pH were optimized first. Then, AuNP was conjugated with anti-DON mAb to synthesize the AuNP-mAb (AuNP probe). The AuNP strip involved four components (sample pad, conjugate pad, absorbent pad, and nitrocellulose (NC) membrane). Goat anti-mouse IgG and DON-KLH coating antigen were sprayed onto the NC to form the control line (C line) and test line (T line), respectively. The AuNP probe was sprayed into the conjugate pad. To evaluate the specificity of the AuNP strip, different toxins including DON, OTA, PA, FB_1_, OA, and AFG1 were, respectively, added into the individual AuNP strip. The visual results were obtained within 10 min. For the sensitivity test, different DON concentrations (4, 5, 6.67, 10, 20, and 50 µg/mL) were dropped into the individual AuNP strip, which was allowed to react with the DON-KLH coating antigen for the limited AuNP probe sprayed into the conjugate pad [33].

### 5.7. Preparation of Gold Nanoflower (AuNF) and AuNF Probe

The seeding growth method was conducted with some modification for synthesis of AuNF [29]. The conventional AuNP with diameter 20 nm was prepared according to the citrate reduction method [40] and used as gold seeds, whereas the HAuCl_4_ solution and sodium hydroquinone citrate were utilized as the growth solution. Briefly, 750 μL of the hydroquinone (1 mol/L), 750 μL of HAuCl_4_ solution (0.01%), and 100 μL of the prepared AuNP solution were placed into 100 mL of distilled water with continuous stirring. After reacting at room temperature for 35 min, the color of the solution was changed to blue, and the morphology of the obtained AuNF solution was characterized by transmission electron microscope (TEM). The preparation of the AuNF probe was conducted as Ji’s method [37] with slight modification. The pH of the AuNF solution as well as the antibody amount was optimized for perfect performance of the AuNF probe. Then the anti-DON mAb was mixed by dropping into the AuNF solution under contentious gentle stirring for 35 min, and the obtained AuNF probe solution was centrifuged at 6000 r/min for 50 min, then suspended in 0.5 mL PBS and placed at 4 °C for further use.

### 5.8. Preparation and Characterization of AuNF-Based Strip

The AuNF strip has four components similar to the conventional AuNP strip. The AuNF probe was sprayed onto the conjugate pad. The goat anti-mouse IgG antibody and DON-KLH conjugate were integrated onto the NC membrane to form the C line and the T line, respectively. To assess specificity of the AuNF strip, different toxins such as DON, AFB_1_, AFB_2_, and AFG_1_ were used to compete with the AuNF probe on the conjugate pad, and the visual result would be determined by the naked eyes within 7–10 min. For sensitivity, 1 mg/mL of the DON toxin standard solution was diluted into suitable concentration (4, 5, 6.67, 10, 20, 50, and 100 µg/mL), and the sensitivity of the test strip was measured according to the experimental results (PBS as negative control).

### 5.9. Samples Detection

Real samples including rice, wheat, and corn samples received from a local market were assayed by the above established AuNP/AuNF -based strip tests. Briefly, 1 g of each sample was crushed and placed into a 50 mL centrifuge tube, then extracted with 2 mL aqueous solution (20% methanol). Then, the mixture was shaken for 25 min and centrifuged at 1000 r/min for 15 min, and the supernatant of each sample was collected and individually dropped into the sample pad of the constructed AuNP/AuNF -based strips, respectively.

### 5.10. Ethical Clearance and Animal Handle

All the animal studies adhered to the Animal Ethics Committee of the Fujian Agriculture and Forestry University, China (C1017/23.Dec.2014). The mice (female Balb/C) were handled under an appropriate condition. The room temperature was set at 25 ± 1 °C with humidity of 47–55%. The total health monitoring of the experimental mice was achieved regularly, and the mice room and cage were cleaned on a regular program.

## Figures and Tables

**Figure 1 toxins-14-00533-f001:**
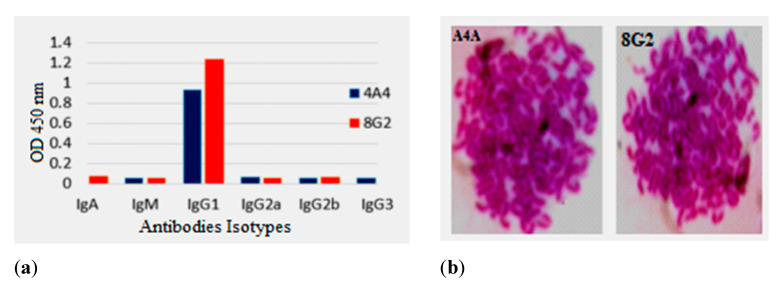
Isotype and chromosome assay: (**a**) Isotype assay of the hybridoma cells 4A4 and 8G2; (**b**) chromosome analysis of the hybridoma cells 4A4 and 8G2.

**Figure 2 toxins-14-00533-f002:**
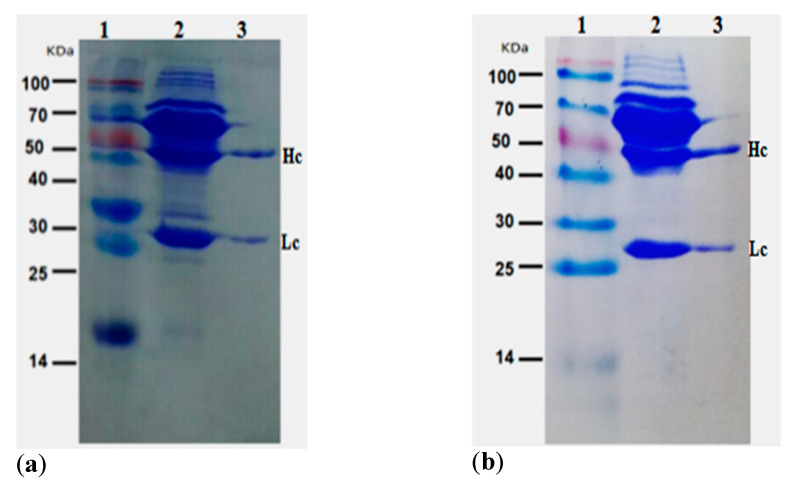
Purification and titer assay of ascites and the purified anti-DON mAb: (**a**) SDS-PAGE assay of ascites and the purified mAb for 4A4; (**b**) SDS-PAGE assay of ascites and the purified mAb for 8G2. lane 1: marker; lane 2: ascites; lane 3: purified antibody; Hc = heavy chain; Lc = light chain; (**c**) titer assay of ascites and purified mAb (4A4 and 8G2).

**Figure 3 toxins-14-00533-f003:**
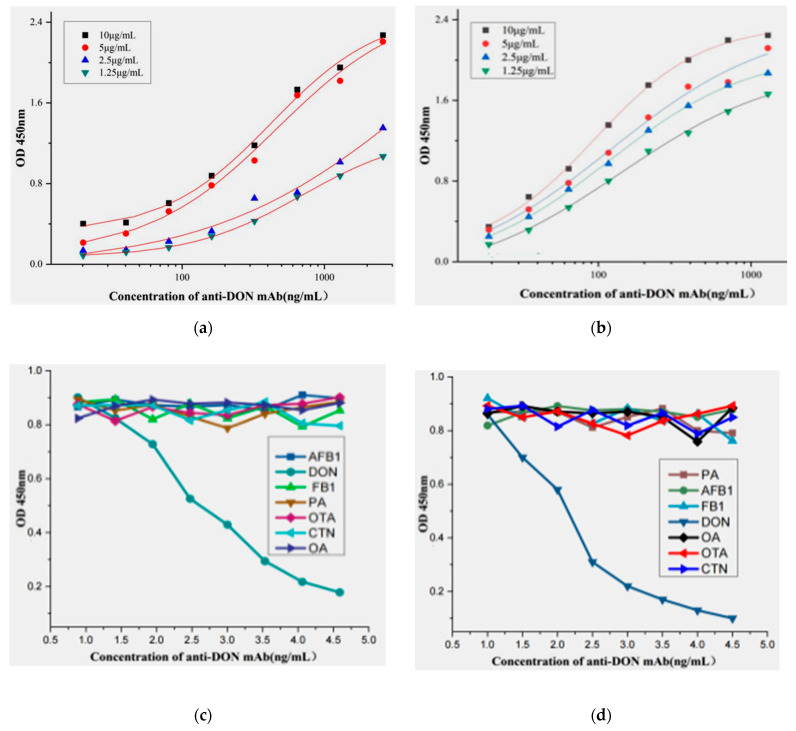
Affinity and cross-reactivity analysis of anti-DON mAb: (**a**) affinity determination of anti-DON mAb from 4A4; (**b**) affinity determination of anti-DON mAb from 8G2; (**c**) specificity of anti-DON mAb from 4A4; (**d**) specificity of anti-DON mAb from 8G2.

**Figure 4 toxins-14-00533-f004:**
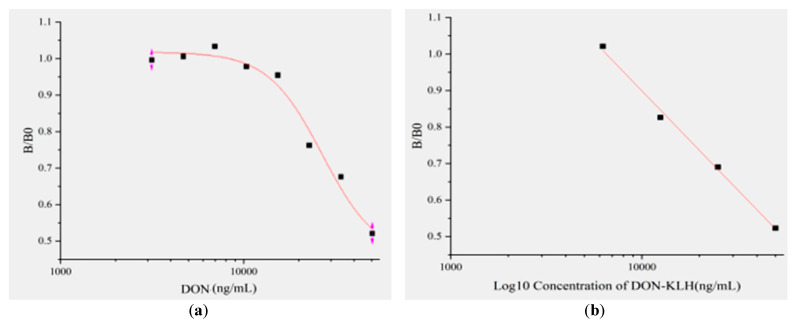
Development of ic-ELISA: (**a**) calibration curve of (B/B0) against DON concentration; (**b**) standard curve for DON determination by ic-ELISA.

**Figure 5 toxins-14-00533-f005:**
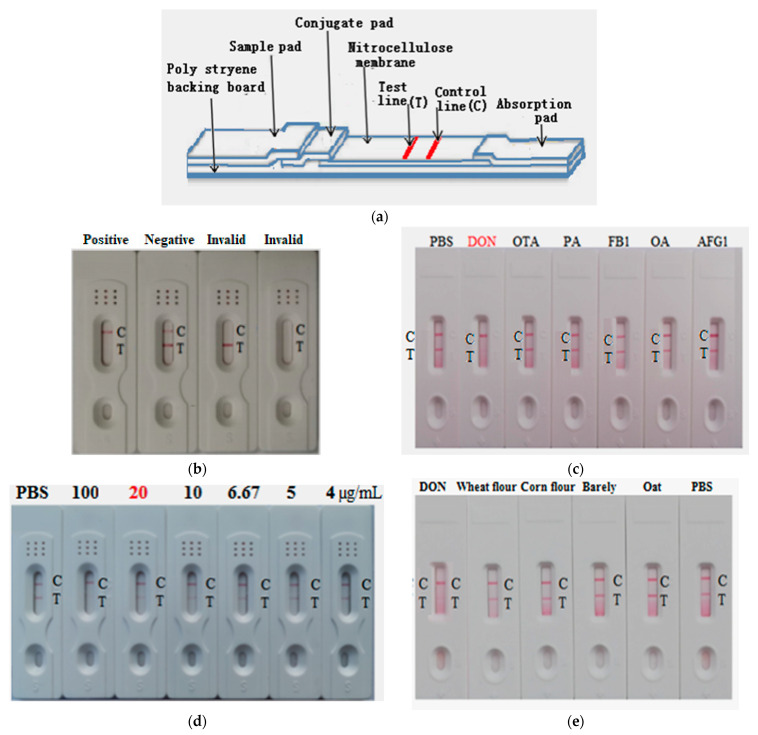
Construction of the AuNP-based strip for determination of DON: (**a**) AuNP strip model; (**b**) construction of the AuNP-based strip; (**c**) specificity of the AuNP-based strip; (**d**) sensitivity of the developed AuNP strip for DON toxin detection; (**e**) real samples were determined by the developed AuNP-based strip.

**Figure 6 toxins-14-00533-f006:**
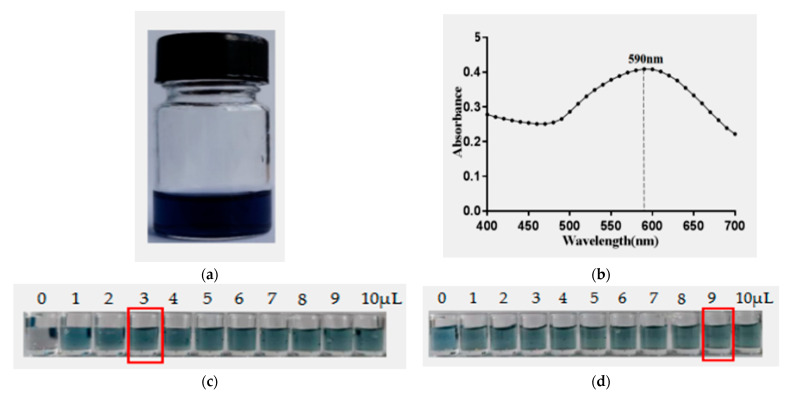
Preparation of the AuNF solution and the AuNF probe: (**a**) preparation of the AuNF solution; (**b**) the spectra of AuNF solution; (**c**) the optimal amount of anti-DON mAb for preparation of AuNF probe; (**d**) optimal volume of K_2_CO_3_ for preparation of the AuNF probe.

**Figure 7 toxins-14-00533-f007:**
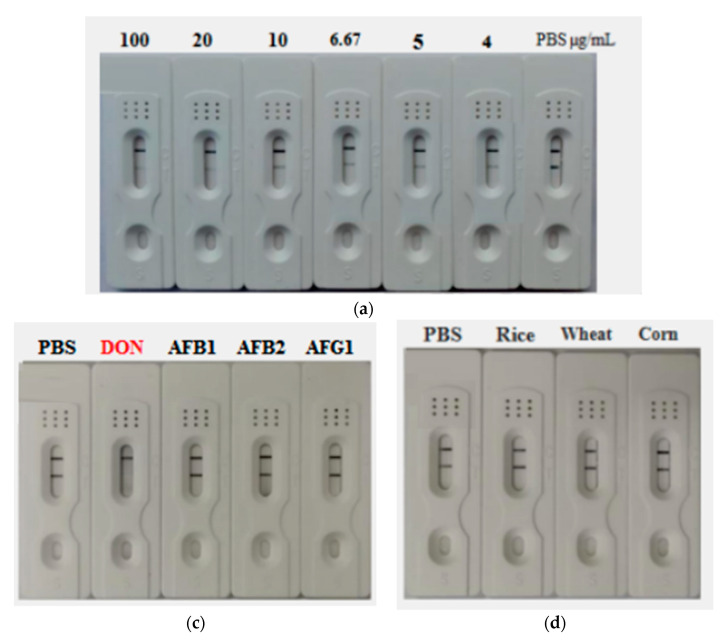
Construction and characterization of the AuNF strip: (**a**) sensitivity of the developed AuNF strip for DON toxin detection; (**b**) specificity of the AuNF strip; (**c**) detection of DON in real samples using the AuNF strip.

**Table 1 toxins-14-00533-t001:** Analysis of DON in spiked corn samples using the developed ic-ELISA (*n* = 3).

Intra Assay(*n* = 3) *						Inter Assay(*n* = 3) #			
Sample	Spiked Level (ng/mL)	Measured Level (ng/mL)	Recovery (%)	SD	CV (%)	Measured Level (ng/mL)	Recovery (%)	SD	CV (%)
Corn	0.5	0.45	90.00	5.342	11. 87	0.46	92	4.54	9.88
	5	4.39	87.80	58.43	13.31	4.77	95.4	34.7	7.27
	100	96.53	96.53	330.1	3.42	96.61	96.61	308.2	3.19
	1000	951.2	91.52	3367.248	3.54	928.4	92.84	13.76	11.02
	Average		91.46				94.21		

Standard deviation (SD); coefficient of variation (CV); (*n* = 3) * within plate in one day; (*n* = 3) # between run in 6 days.

## Data Availability

Not applicable.

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
