# Peer review of "Preparation of Monoclonal Antibody against Deoxynivalenol and Development of Immunoassays"

_toxins, 2022, doi:10.3390/toxins14080533_

Round 1

Reviewer 1 Report

This manuscript describes the development of another monoclonal antibody for the mycotoxin, deoxynivalenol (DON) as a rapid method for assessing agricultural products for DON contamination. The authors describe the development of the antibody and an improvement on sensitivity by using colloidal gold nanoparticles to develop a strip test. It is relatively short at 12 pages, of which 3 are detailed materials and methods but the experimental design is somewhat flawed. The authors fail to justify the requirement for yet another ELISA antibody for DON and why theirs represents an improvement over what has been already published. While the introduction presents other methods for analysis of DON as well as ELISA and lateral flow tests for other toxins, there is no description of what is already in the literature concerning DON ELISA assays. There are many commercial and non-commercial antibodies available (for comparison see Nguyen et al., World Mycotoxin Journal (2019) 12(1), 43-53.) with various sensitivities and cross reactivity with other molecules closely related to DON. In addition, there is no mention in the discussion of why this DON mAb is an improvement over existing ones in terms of quantitation or that the lateral strips developed with this mAb, which are not quantitative, represent an improvement over commercial ELISA methods. The comparison of affinity and specificity of the mAb to other mycotoxins of unrelated structures makes no sense since the mAb was developed using a DON-BSA conjugate. It would have made more sense to check cross reactivity to DON related structures such as 3-ADON, 15-ADON, DON-3G or NX, but that was not reported. The description of the isolation of the new mAb, establishment of standard curve (Figure 4) and recoveries (Table 1) as well as most of the experimental is adequate but there are many grammatical/sentence structure errors throughout the manuscript and often the terms are not defined until the experimental which makes the manuscript difficult to read. References and how/where they are quoted in the text also need to be checked.

Specific comments:

Page 1, lines 5-6; 27; 44-45; 47 – poor grammar and sentence structure, line 10 – define “ic-ELISA”

Page 2, line 51, what is the difference between colloidal gold particle (AuNP) and gold nano-flower (AuNF) and why is it called that?

Page 2, line 52: reference 21 has nothing to do with AuNF – it is an LC-MS study

Page 3, section 2.4: As stated above, testing the mAb against other molecules of completely different structures makes no sense since the mAb was generated from a DON-BSA conjugate. There appears to be no testing against DON-related structures.

Author Response

 Dear Sir,
Thank you for your comments and suggestions, I will take your comment and suggestions point by point which makes my manuscript better and guide me to finish my paper publication.

Author Response

Dear Sir,
Thank you for your comments and suggestions, and   I will take your comment and suggestions point by point makes my manuscript better and guide me to finish my paper publication.

Reviewer 3 Report

Line 22: Fusarium in italic format

Line 27: add extra space

Line 29: reformulated the sentence

Line 31-34: full name of equipment, the first letter in upper case in all names in normal letter

Figure 1a provide the legend of axes xx,yy

Figure 2ab are overlapping, please, correct it

Figure 3 and 4: uniformize the legend of xx, yy axes, type letter and format

Table 1: please, check the instructions for the authors regarding the tables format

Improve results discussion

Line 306, 347: add a space before the units

Line 307, 357, 364: remove extra space

Please revised, carefully, all the manuscript, several sentences have extra spaces, uniformized the format units after the results or results space and units.

Author Response

(The authors gave the same response as above.)

Round 2

Reviewer 1 Report

The authors have responded to two reviewers with quite different criticisms and the manuscript is much improved, especially the introduction and discussion.

Page 4: The testing of the anti-DON MAB against other mycotoxins of completely different structures (there is not even another trichothecene) simply because there are “no DON related toxins in the lab these days” does not necessarily show that the MAB shows high specificity to DON. However, accepted if the editor agrees.

Page 1, line 25: “species” (plural)

Page 1, lines 37 and 38: “highly sensitive” and sophisticated “instrumentation” 

Author Response

Dear sir,  "Please see the attachment." 
